# Neural Network Aided Homogenization Approach for Predicting Effective Thermal Conductivity of Composite Construction Materials

**DOI:** 10.3390/ma16093322

**Published:** 2023-04-23

**Authors:** Zhu Shi, Wenyao Peng, Chaoqun Xiang, Liang Li, Qibin Xie

**Affiliations:** 1Hunan Road and Bridge Construction Group Co., Ltd., Changsha 410075, China; zhushi_hcig@163.com (Z.S.); xiangcq888@163.com (C.X.); 2Hunan Pingyi Expressway Construction and Development Co., Ltd., Changsha 410004, China; wypeng_hcig@163.com; 3Department of Civil Engineering, Central South University, Changsha 410075, China; xieqibin@csu.edu.cn

**Keywords:** effective thermal conductivity, neural network, homogenization, asphalt mixture, cement concrete

## Abstract

Thermal conductivity is a fundamental material parameter involved in various infrastructure design guides around the world. This paper developed an innovative neural network (NN) aided homogenization approach for predicting the effective thermal conductivity of various composite construction materials. The 2-D meso-structures of dense graded asphalt mixture, porous asphalt mixture, and cement concrete were generated and divided into 2^n^ × 2^n^ square elements with specific thermal conductivity values. A two-layer feed-forward neural network with sigmoid hidden neurons and linear output neurons was built to predict the effective thermal conductivity of the 2 × 2 block. The Levenberg-Marquardt backpropagation algorithm was used to train the network. By repeatedly using the neural network, the effective thermal conductivities of 2-D meso-structures were calculated. The accuracy of the above NN aided homogenization approach was validated with experiment, and various factors affecting the effective thermal conductivity were analyzed. The analysis results show that the accuracy of the NN aided approach is acceptable with relative errors of 1.92~4.34% for the dense graded asphalt mixture, 1.10~6.85% for the porous asphalt mixture, and 1.13~3.14% for the cement concrete. The relative errors for all the materials are lower than 5% when the heterogeneous structures are divided into 512 × 512 elements. Ignoring the actual material meso-structures may lead to significant errors (134.01%) in predicting the effective thermal conductivity of materials with high heterogeneity such as porous asphalt mixture. While proper simplification is acceptable for dense construction composite materials. The effective thermal conductivity of composite cement-asphalt mixtures increases with higher saturation of grouted material. However, the improvement effect of the high-conductive cement paste on the composite cement-asphalt mixtures could be significantly reduced when the cement paste concentrates at the bottom of the mixture. Cracked aggregates and segregation of material components tend to decrease the effective thermal conductivity of construction materials. The NN aided homogenization approach presented in this paper is useful for selecting the effective thermal conductivity of construction materials.

## 1. Introduction

Frequent fluctuations in temperatures not only increase the risk of thermal cracking in the infrastructures, but also affect the energy consumption [1,2]. Since thermal conductivity of construction material is a critical factor that affects the temperature field of infrastructures in the natural environment, this important material property has been considered in various design guides and standards for infrastructures. The American Concrete Institute (ACI) highlights the importance of concrete thermal properties for energy-conserving building design [3]. The Mechanistic—Empirical Pavement Design Guide (MEPDG) takes thermal conductivity as a parameter that must be considered in pavement design [4]. The typical values of thermal conductivity usually range from 0.5 to 2.5 W/mK for asphalt mixture, and from 0.5 to 4.0 W/mK for cement concrete [5]. In general, the effective thermal conductivity of construction materials is affected by raw material properties and composition and may change with time due to material aging [6]. However, due to the lack of a clear basis, choosing thermal conductivity from the typical range tends to be subjective and not reliable enough for design. Moreover, for construction materials containing special components such as reclaimed asphalt and rejuvenators [7], the effective thermal conductivity may exceed the typical range. Therefore, a more accurate method for determining the effective thermal conductivity is needed.

At present, laboratory testing is the most common method for determining the thermal conductivity of cement and asphalt composite materials when high precision is required. Most existing experimental studies on thermal conductivity of construction materials were based on either the steady-state heat transfer method [8,9] or the transient heat transfer method [10,11,12,13,14]. Since the steady-state method requires the temperature field of the specimen to be stable, its time consumption is usually high, especially for construction materials with low conductivity. On the other hand, in the transient heat transfer method, a thin plane heat source is clamped by two pieces of tested samples. The thermal conductivity of the samples is determined by analyzing the temperature profile. Due to its high efficiency, the transient heat transfer method has been widely used to test the thermal conductivity of various cement-based and asphalt-based construction materials. However, since the thin heat source is relatively small, the accuracy of the transient heat transfer method may be affected when the sample is highly heterogeneous.

To avoid the limitations of laboratory testing, various models have been developed to predict the effective thermal conductivity of composite construction materials based on the material composition. The basic idea of these models is that the macro material properties are determined by the meso-structure of the material. When the structure of the material is simplified to an idealized or periodic form, the explicit solution of the effective thermal conductivity can be derived, such as with the series model and parallel model. However, since the material structures of construction materials are usually more complex than the above idealized ones, simply applying these models on construction materials may cause large errors. To solve this problem, one approach is to develop numerical models based on heterogeneous structures [15,16,17,18,19,20,21,22,23], and the other is to derive more complex but applicable analytical or semi-analytical solutions [24,25,26,27,28,29]. In the numerical models, the effective thermal conductivity was calculated by analyzing the heat flux in the representative heterogeneous structure in the steady state. In theory, as long as the heterogeneous structure being used in these finite element models are reasonable, these models could provide relatively accurate results. However, the computational costs of the finite element models are usually high. Compared with the numerical models, the analytical or semi-analytical models are usually more efficient. A commonly used strategy is to define a representative multiphase system containing discrete inclusions and a continuous matrix. The effective thermal conductivities of these multiphase systems were derived, and then used to predict the effective thermal conductivity of the whole mixture via multiscale homogenization [25,26]. The analytical models establish the relationship between the meso-structure and the macroscopic thermal conductivity of materials. When the meso-structure of material is too complex, it is difficult to determine the above relationship. Neural network is a powerful tool to find the relationship between various variables under complex conditions and may provide an alternative way to predict the effective thermal conductivity of composite construction materials based on their specific heterogeneous structures [30]. Lee et al. [31] and Sargam et al. [32] developed neural network models to predict the effective thermal conductivity of concrete. These models considered factors related to material composition and properties but ignored the effect of material meso-structure. Therefore, these models cannot provide predictions based on the meso-structure of construction materials like asphalt mixture and cement concrete. The present study aims to develop a model for predicting the effective thermal conductivity of composite construction materials through the neutral network, by using the specific heterogeneous meso-structures of the materials. This model presented in this paper would be useful for selecting effective thermal conductivity of construction materials for infrastructure design and analysis.

## 2. Objectives

The objective of this study is to develop a neural network aided homogenization approach for predicting the effective thermal conductivity of various composite construction materials, including different types of asphalt mixtures and cement concrete. The 2-D meso-structures of these materials were divided into 2^n^ × 2^n^ square elements with specific thermal conductivity values. A two-layer feed-forward neural network with sigmoid hidden neurons and linear output neurons was built to predict the effective thermal conductivity of the 2 × 2 block. By repeatedly using the neural network, the effective thermal conductivities of 2-D meso-structures were calculated. The accuracy of the above neural network aided homogenization approach was validated with experiment, and various factors affecting the effective thermal conductivity were analyzed.

## 3. Methodology

### 3.1. Homogenization Scheme

From a meso-scope, the thermal conductivity of composite construction materials like asphalt mixture and cement concrete is not a constant, but a variable that varies with the position. However, the thermal conductivity used in engineering practice is usually a macroscopic representative value, namely, the effective thermal conductivity. In this paper, the homogenization of thermal conductivity refers to the process of converting the mesoscopic non-uniform thermal conductivity into the macroscopic effective thermal conductivity. Figure 1 shows a schematic diagram of the homogenization scheme. The 2-D meso-structure was divided into 2^n^ × 2^n^ square elements. Each element was assigned a thermal conductivity value based on the material composition at the specific location. The difference in the thermal conductivity value was represented by the grayscale values. For example, the 2^n^ × 2^n^ model shown in Figure 2 only contained two different materials, namely, continuous mortar and discrete aggregates. Thus, the color of these 2^n^ × 2^n^ square elements was either black or white, depending on the thermal conductivity. Then, if the effective thermal conductivity of each 2 × 2 element could be calculated, the 2^n^ × 2^n^ model could be reduced into a 2^n−1^ × 2^n−1^ model. By repeating the above calculation, the 2^n−1^ × 2^n−1^ model could be further reduced into 2^n−2^ × 2^n−2^ model, into a 2^n−3^ × 2^n−3^ model, and finally into a model with a single value. In the above homogenization scheme, the most critical process is determining the relationship between the thermal conductivities of the 2 × 2 block (*k*_1_, *k*_2_, *k*_3_, and *k*_4_) and the effective thermal conductivity *k*_e_. The exact solution for the above problem could be derived when the material structure has certain characteristics. For example, the explicit solutions of conductivity when the material has a periodic checkerboard structure were derived in previous studies [13,14]. However, when the material structure is random and complex, it is not easy to develop a concise explicit solution. Therefore, in this paper, the relationship between 2 × 2 block conductivity and effective thermal conductivity was built with a neural network.

### 3.2. Neural Network Approach

A two-layer feed-forward neural network with sigmoid hidden neurons and linear output neurons was built to predict the effective thermal conductivity *k*_e_ of a 2 × 2 block, as shown in Figure 2. The number of hidden neurons was 20 in the neural network. The network was built and trained in MATLAB, and the Levenberg-Marquardt backpropagation algorithm was used for training.

The dataset used for training and validation of the neural network was obtained by conducting a series of simple finite element analysis. In these finite element analyses, constant temperatures *T*_1_ and *T*_2_ were assigned to the upper and lower boundaries of the 2 × 2 block, respectively, as shown in Figure 3. While the other vertical boundaries were set to be insulated. When heat conduction reached steady state, the effective thermal conductivity of the 2 × 2 block could be calculated with Equation (1).
(1)$$k_{e}=\frac{\sum_{j=1}^{m}q_{j}a_{j}}{B(T_{1}-T_{2})}$$
where, *B* is the side length of the 2 × 2 block (m); *q_j_* is the vertical heat flux in the *j*-th finite element mesh (W/m^2^); *a_j_* is the area of the *j*-th finite element mesh (m^2^).

In Figure 3a, when *k*_1_ = *k*_2_, and *k*_3_ = *k*_4_, the 2 × 2 block could be regarded as a series model. When *k*_1_ = *k*_3_, and *k*_2_ = *k*_4_, the 2 × 2 block could be regarded as a parallel model. In the above two special cases, the effective thermal conductivity could be calculated with analytical equations. Figure 3b shows the comparison between the results from the finite element analysis and analytical equations and proves the accuracy of the finite element analysis in these special cases. Moreover, since Equation (1) is strictly derived from Fourier’s law, and the steady heat transfer mode shown in Figure 2 is quite simple, it is believed that the results from the finite element analysis are credible.

In total, 190,880 sets of data were prepared with the above finite element analysis for developing the neural network. The literature shows that the thermal conductivity of the aggregate is higher than asphalt, cement, and air in most cases, and typical values of commonly used aggregates range from 1~4 W/mK. Therefore, in the above dataset, *k*_1_, *k*_2_, *k*_3_, and *k*_4_ all ranged from 0~5 W/mK. For the 190,880 sets of data, the values of *k*_e_ were calculated in COMSOL in batch. Then the data were randomly divided into three groups. Most of the data (80%) were used for the network training, and the rest were used for the validation (10%) and testing (10%). Figure 4 shows the error histogram of the neural network. For most cases, the absolute error ranged between ±0.03. Moreover, a series of statistical analyses were conducted on the 190,880 sets of predicted values (NN values) and true values (FEM values). Results show that the R-squared (R^2^) is 0.99996, the root mean square error (RMSE) is 0.0067, the mean absolute error (MAE) is 0.0045, the mean squared error (MSE) is 4.51 × 10^−5^, and the covariance (COV) is 1.224. Based on the above results, it can be concluded that the accuracy of the neural network model is generally acceptable.

### 3.3. Material Structure Modelling

In this study, three types of commonly used composite construction materials were considered, including a dense graded asphalt mixture (DA), a porous asphalt mixture (PA), and a cement concrete (CC). The dense graded asphalt mixture was modelled as a composite consisted of coarse aggregates and fine aggregate matrix (FAM-DA), as shown in Figure 5a. The FAM-DA was a mixture composed of asphalt, air voids, and fine particles smaller than 2.36 mm. The porous asphalt mixture was modelled as a composite consisted of coarse aggregates, pores, and fine aggregate matrix (FAM-PA), as shown in Figure 5b. The FAM-PA was a mixture composed of asphalt and fine particles smaller than 2.36 mm. The cement concrete was modelled as a composite and consisted of coarse aggregates and fine aggregate matrix (FAM-CC), as shown in Figure 5c. The FAM-CC was a mixture composed of cement mortar with fine particles smaller than 4.75 mm. The digital specimen size for each material shown in Figure 5 was 12.8 cm.

To generate the material structures shown in Figure 5, the Aggregate Image System (AIMS) was used to capture the 2-D images of different sized aggregates. For the dense graded asphalt mixture (DA) and the cement concrete (CC), the 2-D images of aggregates were randomly placed into the FAM-DA and FAM-CC, respectively. While for the porous asphalt mixture (PA), the aggregates were wrapped with a film of FAM-PA, and then placed into the continuous medium of pores. After all the aggregates were placed, the material structures shown in Figure 5 were converted into 2^n^ × 2^n^ square elements, which could be used in the neural network aided homogenization described in previous sections. Since the value of *n* may affect the accuracy and computational cost of the overall homogenization, this value should be determined properly. In this paper, the value of *n* for most cases was taken as 9 based on a series of parameter analysis, which will be presented in the results and discussion section of this paper.

## 4. Experiment

### 4.1. Materials

A series of experiments were conducted to measure the effective thermal conductivity of different types of construction material and validate the neural network aided homogenization approach presented in this paper. The asphalt used for dense graded and porous asphalt mixture was PG 76-22 asphalt. The cement used for the cement concrete was ordinary Portland cement. Granite aggregates were used as coarse aggregates of asphalt mixtures and cement concrete and were used as fine aggregates of asphalt mixtures. The fine aggregates in the cement mortar were river sand. The designed asphalt content was about 4.3% for the asphalt mixtures. The aimed air void content was 6% for the dense graded asphalt mixture and was 24% for the porous asphalt mixture. The aggregate gradations for the asphalt mixtures and cement concrete were summarized in Table 1. The mix proportion of concrete was shown in Table 2. Figure 6 presented examples of the testing samples.

### 4.2. Test Method and Results

The thermal conductivities of the dense graded asphalt mixture, cement concrete, and fine aggregate matrix (FAM-DA, FAM-PA, FAM-CC) were measured with DRE-2C thermal conductivity tester (Figure 7), based on the transient plane heat source (TPS) method. The thermal conductivities of the air and the granite aggregate were taken as typical values, 0.025 and 2.855 W/mK, respectively [16]. Considering the surface of the porous asphalt mixture is uneven, the effective thermal conductivity of the porous asphalt mixture was measured with the method recommended in a previous study [33]. The test results were shown in Table 3. Table 3 shows that the thermal conductivity of cement concrete (CC) is the largest, while that of porous asphalt mixture (PA) is the smallest. For both cement concrete (CC) and dense graded asphalt mixture (DA), the effective thermal conductivity of the whole mixture is higher than that of the fine aggregate matrix (FAM-CC and FAM-DA). This is because many coarse aggregates are distributed outside the fine aggregate matrix. While for the porous asphalt mixture (PA), the effective thermal conductivity of the whole mixture is lower than that of the fine aggregate matrix (FAM-PA), since most large-scale interconnected pores are outside the fine aggregate matrix.

## 5. Results and Discussion

### 5.1. Model Validation

To validate the neural network model presented in this paper, the effective thermal conductivity of dense graded asphalt mixture, porous asphalt mixture, and cement concrete was predicted with the neural network aided homogenization approach. For each type of materials, five different digital samples were generated with the material structure modelling method described in the previous section. In these digital samples, the material compositions could be kept exactly the same as that used in the experiment. Figure 8a shows the comparison between the predicted and measured values. The predicted average effective thermal conductivities of dense graded asphalt mixture (DA), porous asphalt mixture (PA), and cement concrete (CC) were 2.070, 0.444, and 2.161 W/mK, respectively. While the measured values of dense graded asphalt mixture (DA), porous asphalt mixture (PA), and cement concrete (CC) were 2.127, 0.429, and 2.213 W/mK, respectively. In general, the accuracy of the neural network model presented in this paper is acceptable.

A more detailed analysis on the relative errors between the predicted and measured values are shown in Figure 8b. Results show that the relative errors range from 1.92~4.34% for the dense graded asphalt mixture, from 1.10~6.85% for the porous asphalt mixture, and from 1.13~3.14% for the cement concrete. The average relative error for the porous asphalt mixture is the highest. One possible reason is that the air void content in the porous asphalt mixture is much higher. Since the thermal conductivity of the air is much lower than that of the asphalt and aggregate [5,16], the heat flux inside the mixture tends to transfer through solid phase (asphalt and aggregates). Thus, the connected pore structures block heat flux and make the heat transfer inside the porous asphalt mixture more complex. As a result, it is more challenging to ensure the prediction accuracy for the porous asphalt mixture.

### 5.2. Discussion on the Number of Elements

In the neural network aided homogenization approach presented in this paper, the heterogeneous structure of material was discretized into 2^n^ × 2^n^ square elements. For a given specimen size, a smaller value of *n* may reduce the computational cost but affect the accuracy. On the contrary, a larger value of *n* may improve the accuracy but affect the computational efficiency. In order to investigate the effect of the number of elements on the results, different values of *n* (7, 8, and 9) were used to generate the material structures. In other words, the material structures were discretized into 128 × 128, 256 × 256, and 512 × 512 elements, respectively. The effective thermal conductivity of different types of construction materials were predicted with the neural network aided homogenization approach, and the relative errors between the predicted and measured values were calculated. Figure 9 shows the relative error for the dense graded asphalt mixture (DA), porous asphalt mixture (PA), and cement concrete (CC). It is observed that in either case, the relative error for PA is the largest. This is understandable since the heterogeneity in PA is the most significant. When a smaller value of *n* is applied, the number of pixels in the image will be reduced, which will lead to inaccurate characterization of the material structures. As shown in Figure 9, when *n* = 9, the relative errors for all the materials are lower than 5%. Therefore, in this paper, the value of *n* was taken as 9 for model validation and analysis.

### 5.3. Discussion on Material Meso-Structure

As composite materials, the meso-structures of asphalt mixtures and cement concrete should affect the macro thermal properties of the materials. This is an important reason why many studies [15,16,17,18,19,20,21,22], including this one, used models that closely resemble the structure of real asphalt mixtures and cement concrete for analysis. However, it is also reported in the literature that exact microstructure representation of these materials is unnecessary for thermal conductivity prediction [34,35]. In this section, the effect of material meso-structure on the effective thermal conductivity was analyzed. Two different types of heterogeneous structure were generated, and marked as type S and type R. For type S, heterogeneous structures that were similar to the real construction materials were generated. This type of structure is actually that used in the model validation and is shown in Figure 5. While for type R, heterogeneous structures were generated totally randomly, considering the volume fraction of each component only. A schematic diagram of type R is shown in Figure 10. The volume fraction of each component for these materials was the same with that used in the model validation.

The effective thermal conductivities of different materials were calculated with the neural network aided homogenization approach, based on type S structure (Figure 5) and type R structure (Figure 10). The results were shown in Figure 11, together with the measured thermal conductivity. For the dense graded asphalt mixture (DA) and the cement concrete (CC), there is no obvious difference between the results based on type S structure and type R structure. Additionally, the results based on type S structure and type R structure both match well with the experimental data. However, for the porous asphalt mixture (PA), the results based on the type R structure are much different from the experimental data, with an average relative error of 134.01%. This finding indicates that the effective thermal conductivity of the dense graded asphalt mixture and cement concrete can be accurately evaluated with a simplified random structure like type R, but the effective thermal conductivity of porous asphalt mixture cannot. The authors believe that this phenomenon can be explained from the view of material meso-structure. In the type S porous asphalt mixture shown in Figure 5b, there are many large-scale interconnected pores. Since the thermal conductivity of the air is much lower than that of asphalt and aggregate, the heat flux tends to flow along the path with the higher thermal conductivity, that is, the path linked by aggregates. In this case, these large-scale interconnected pores could significantly reduce the heat transfer rate by cutting off the heat transfer path. However, in the type R porous asphalt mixture shown in Figure 10b, the pores, aggregates, and fine aggregate matrix are randomly distributed. The interconnected pores were much fewer than that in the type S porous asphalt mixture. Compared with large-scale interconnected pores, it is easier for the heat flux to transfer around these smaller air voids. As a result, the simplified random structure (type R) fails to adequately reflect the blocking effect of interconnected pores on heat flow, thus overestimating the effective thermal conductivity of the porous asphalt mixture. While for the dense graded asphalt mixture and cement concrete, although the material compositions are different, both these two types of materials are dense without large-scale, high-volume pores. That means the block effect of air to the heat transfer would be very limited. The above analysis implies that although it is acceptable to simplify dense construction composite materials as random blocks when predicting the thermal conductivity, exact heterogeneous structure modelling is still necessary, especially for materials with high heterogeneity such as porous asphalt mixture.

### 5.4. Discussion on Thermal Conductivity Improvement

The thermal conductivity of construction materials is a critical factor that affects the heat exchange between the infrastructures and the external environment. When higher heat transfer efficiency is needed, the thermal conductivities of construction materials need to be improved. For the porous asphalt mixture with relatively low thermal conductivity, a feasible method is grouting high-conductive cement paste into the pore structures inside the porous asphalt mixture. This type of material is usually called composite cement-asphalt mixture, cement grout asphalt composite, or semiflexible mixture [36,37]. However, due to limited fluidity of the cement paste and connectivity of pores, it is possible that only part of the pores is filled with cement paste. For example, for the porous asphalt mixture shown in Figure 12a, the ideal condition is to fill all the pores with cement paste, namely, a condition with saturation of 100%, as shown in Figure 12b. However, when only a part of the pores is filled with cement paste, the saturation of cement paste is lower than 100%. Figure 12c,d show two samples with saturations of 20% and 60%, respectively. In these two samples, the grouted cement paste concentrates at the bottom of the samples. While in the samples shown in Figure 12e,f, the cement paste concentrates at one side of the samples.

In order to investigate the improvement effect of high-conductive cement paste on the composite cement-asphalt mixtures, digital specimens with different saturations were generated. The high-conductive cement paste was either horizontally distributed at the bottom of the specimen (Figure 12c,d), or vertically distributed at one side of the specimen (Figure 12e,f). The thermal conductivity of the high-conductive cement paste was taken as 1.42 and 2.76 W/mK based on the existing literature [38]. The effective thermal conductivity of the composite cement-asphalt mixtures was calculated with the neural network aided homogenization approach presented in this paper. The results are shown in Figure 13.

As shown in Figure 13, the effective thermal conductivity of composite cement-asphalt mixtures increased with higher saturation of grouted material. Moreover, it is observed that the growth rates of effective thermal conductivity depend on the distribution patterns of the high-conductive cement paste. As shown in Figure 13a, when the high-conductive cement paste is concentrated at the bottom of the specimen, the effective thermal conductivity of the composite cement-asphalt mixture increases slowly with the increase of saturation until the saturation reaches 80%. Then a significant jump of effective thermal conductivity is observed when saturation increases from 80% to 100%. This phenomenon is not observed when the high-conductive cement paste is vertically distributed at one side of the specimen. Instead, in this case, the effective thermal conductivity of composite cement-asphalt mixture is roughly linearly related to the saturation. In addition, except for the cases when saturation is 0 or 100%, the effective thermal conductivity of composite cement-asphalt mixture with vertically distributed cement paste is always larger than that with horizontally distributed cement paste. This is because the vertically distributed high-conductive cement paste provides a fast channel for heat transfer in the vertical direction. The above analysis indicates that the thermal conductivity of the grouted cement paste is not the only factor affecting the overall thermal conductivity of the composite cement-asphalt mixture. If the interconnected pores are not fully filled, the improvement effect of high-conductive cement paste on the composite cement-asphalt mixtures will be significantly reduced, especially when the cement paste concentrates at the bottom of the mixture.

### 5.5. Effect of Aggregate Cracking on Thermal Conductivity

The dense graded asphalt mixture is a commonly used construction material for pavement structure. When the asphalt overlay is over compacted during construction, some aggregates could be crushed. In this section, some digital samples of dense graded asphalt mixture containing crushed aggregates were generated. Figure 14 shows an example of dense graded asphalt mixture containing eleven crushed aggregates. The width of the cracks was kept to 0.5 mm, and the thermal conductivity of the crack was set to be the same as the air. To avoid the interference of crack orientation (horizontal or vertical), the crack orientation was set to be 45° for all the cracks.

With the neural network aided homogenization approach presented in this paper, the effective thermal conductivity of dense graded asphalt mixture with 0, 5, 10, 15, 20, and 25 cracked aggregates was calculated. For each case, four digital specimens were generated and analyzed. The results are shown as boxplot in Figure 15. It is observed that the effective thermal conductivity of dense graded asphalt mixture gradually decreased with the increasing content of cracked aggregates. This phenomenon could be explained from two aspects. First, from the view of heat transfer, the cracks could be regarded as air with relatively low thermal conductivity. As a result, the appearance of cracks reduced the overall thermal conductivity of the aggregates. Second, from the view of molecules, as a previous study pointed out, if the molecules are far from each other, the conductivity will be low due to the high energy needed when the molecules vibrate [39]. Similarly, for an intact rock, the molecules are close to each other. While for a cracked rock, some of the molecules are isolated from each other by cracks. Therefore, the effective thermal conductivity of dense graded asphalt mixture gradually decreased with increasing content of cracked aggregates.

### 5.6. Effect of Concrete Segregation on Thermal Conductivity

The segregation of concrete components is a complex phenomenon and has a negative impact on the mechanical properties of concrete [40,41]. In this section, the effect of segregation on the effective thermal conductivity of cement concrete was investigated with the approach presented in this paper. Three concrete specimens with the same material composition, but different segregation conditions were generated, as shown in Figure 16. The specimen in condition A is concrete without segregation, and the aggregates are uniformly distributed in the whole specimen. The specimens in condition B and condition C represent concrete with the segregation. No aggregate is distributed in the upper 10% and upper 20% of the specimens in condition B and condition C, respectively.

The effective thermal conductivity of the specimens shown in Figure 16 was calculated. Results show that segregation of concrete components tends to decrease the effective thermal conductivity of cement concrete, as shown in Figure 17. This finding is similar with the results in Section 5.4. That is, layered structure perpendicular to the heat flow direction could block the heat transfer. Although the effective thermal conductivity in the aggregate concentrated part may be increased, the overall thermal conductivity of the cement concrete is decreased.

### 5.7. Summary of Discussion

The analysis and discussion in the previous sections proved the effectiveness of the NN aided homogenization approach and presented some interesting findings. In general, the number of elements could significantly affect the calculation results. For a given specimen size, fewer elements may reduce the computational cost, but affect the accuracy. On the contrary, more elements may improve the accuracy, but affect the computational efficiency. Including the effect of realistic meso-structures is important for predicting the effective thermal conductivity of composite construction materials, especially for the porous material. Since the improvement effect of high-conductive cement paste could be reduced when the pores were not fully filled, it is necessary to pay attention to the fill rate in engineering practice. Cracked aggregates and segregation of material components tend to decrease the effective thermal conductivity of construction materials, which indicates that the construction process could affect the thermal behavior of infrastructure.

It should be noted that the NN aided homogenization approach presented in this study can only predict effective thermal conductivity. This is a potential limitation of this approach because it cannot be used to calculate heat flux and temperature fields inside the specimens. It is recommended to involve heat flux and temperature fields in future studies.

## 6. Conclusions

This paper developed a neural network aided homogenization approach for predicting the effective thermal conductivity of various composite construction materials, including different types of asphalt mixtures and cement concrete. The accuracy of the above approach was validated with experiment, and various factors affecting the effective thermal conductivity were analyzed. The NN aided homogenization approach presented in this paper is useful for selecting the effective thermal conductivity of construction materials for infrastructure design and analysis. The following conclusions were concluded from the analysis:(1)The accuracy of the neural network aided approach is acceptable with relative errors of 1.92~4.34% for the dense graded asphalt mixture, 1.10~6.85% for the porous asphalt mixture, and 1.13~3.14% for the cement concrete.(2)For a given heterogeneous structure discretized into 2^n^ × 2^n^ square elements, when *n* = 9, the relative errors for all the materials are lower than 5%, indicating it is reasonable to divide the heterogeneous structure into 512 × 512 elements.(3)Ignoring the actual material meso-structures may lead to significant errors (134.01%) in predicting the effective thermal conductivity of materials with high heterogeneity such as porous asphalt mixture. However, proper simplification is acceptable for dense construction composite materials.(4)The effective thermal conductivity of composite cement-asphalt mixtures increases with the higher saturation of grouted material. However, the improvement effect of a high-conductive cement paste on the composite cement-asphalt mixtures could be significantly reduced when the cement paste concentrates at the bottom of the mixture.(5)Cracked aggregates may slightly reduce the effective thermal conductivity of dense graded asphalt mixture.(6)Segregation of concrete components tends to decrease the effective thermal conductivity of cement concrete.

## Figures and Tables

**Figure 1 materials-16-03322-f001:**
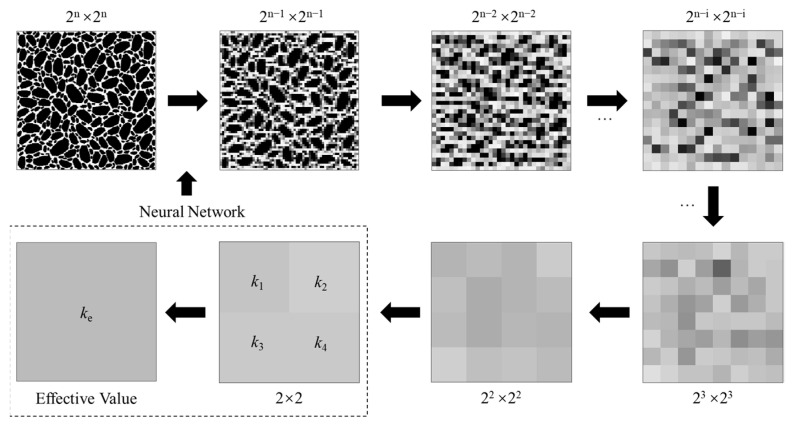
Schematic diagram of the homogenization scheme.

**Figure 2 materials-16-03322-f002:**
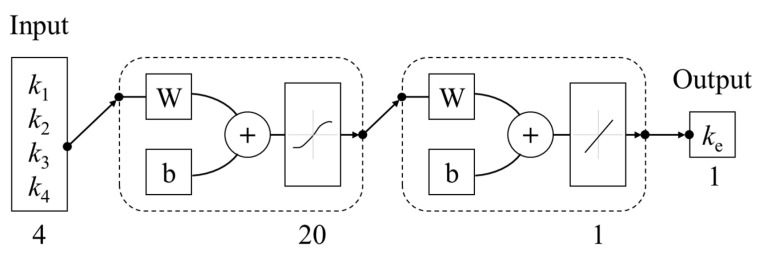
Schematic diagram of neural network model.

**Figure 3 materials-16-03322-f003:**
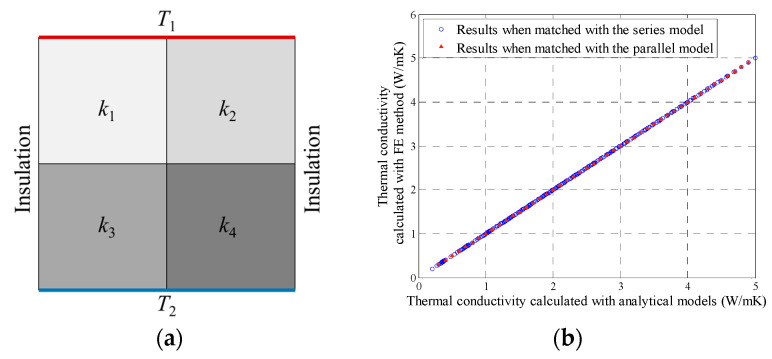
Finite element model and validation (**a**) schematic diagram (**b**) validation for special cases.

**Figure 4 materials-16-03322-f004:**
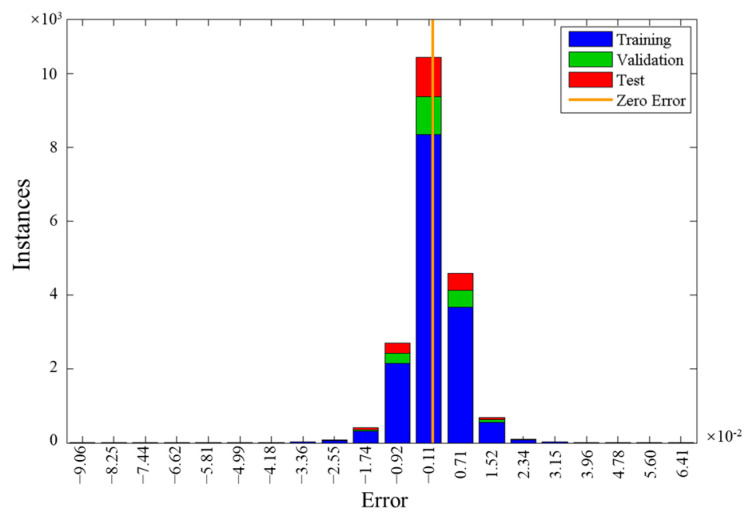
Error histogram of the neural network.

**Figure 5 materials-16-03322-f005:**
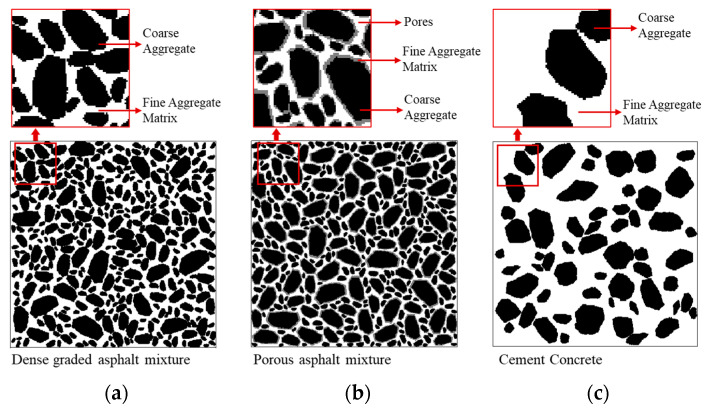
Examples of generated material structure (**a**) dense graded asphalt mixture (**b**) porous asphalt mixture and (**c**) cement concrete.

**Figure 6 materials-16-03322-f006:**
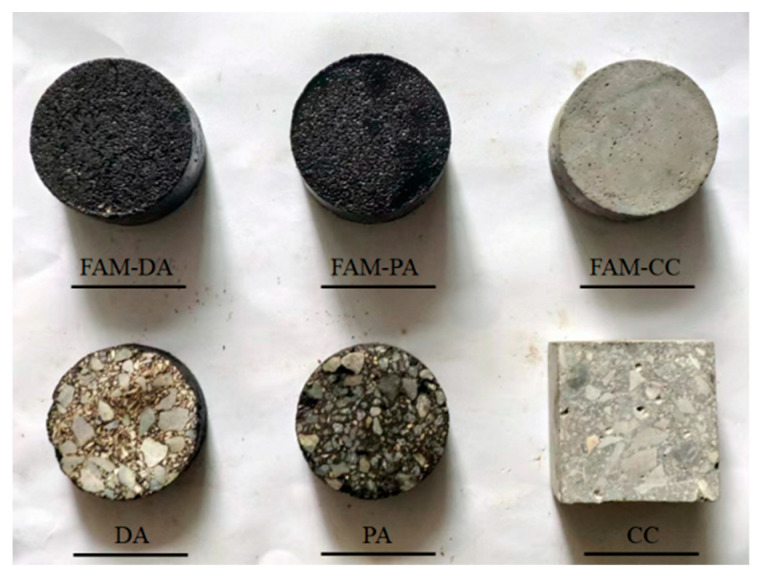
Examples of the testing samples.

**Figure 7 materials-16-03322-f007:**
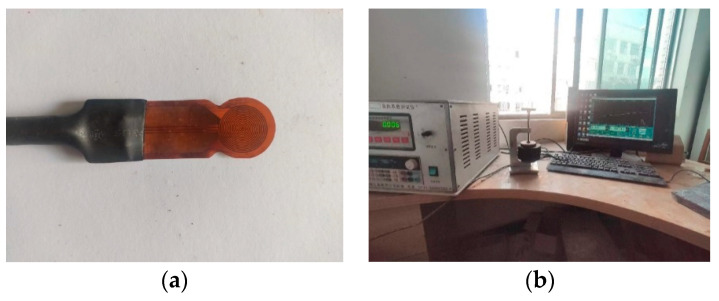
DRE-2C thermal conductivity tester (**a**) sensor, (**b**) equipment.

**Figure 8 materials-16-03322-f008:**
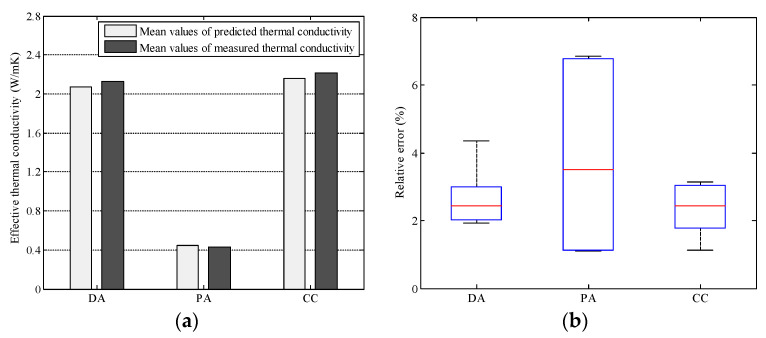
Validation of the neural network model (**a**) comparison between the predicted and measured values, and (**b**) relative error.

**Figure 9 materials-16-03322-f009:**
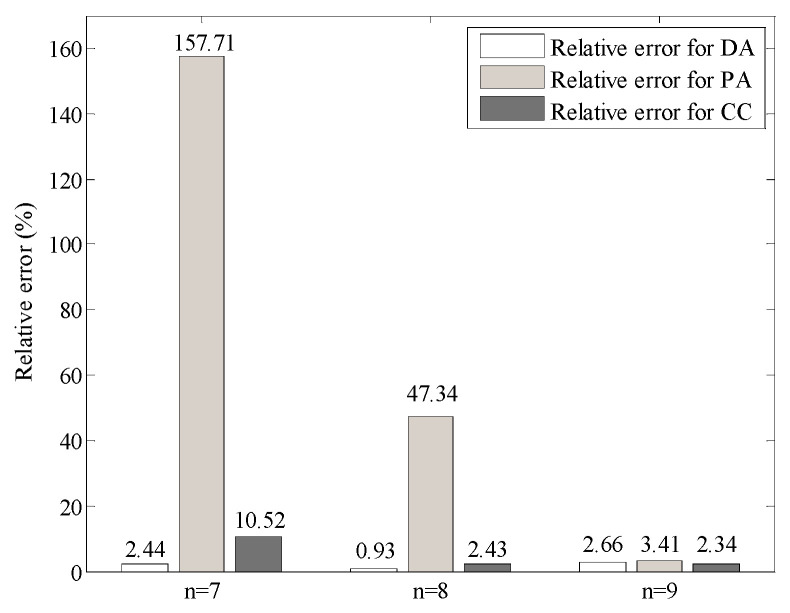
Relative error with different number of elements.

**Figure 10 materials-16-03322-f010:**
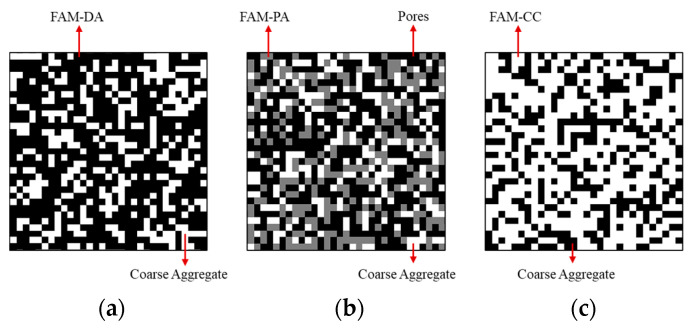
Schematic diagram of random structure type R (**a**) dense graded asphalt mixture, (**b**) porous asphalt mixture, and (**c**) cement concrete.

**Figure 11 materials-16-03322-f011:**
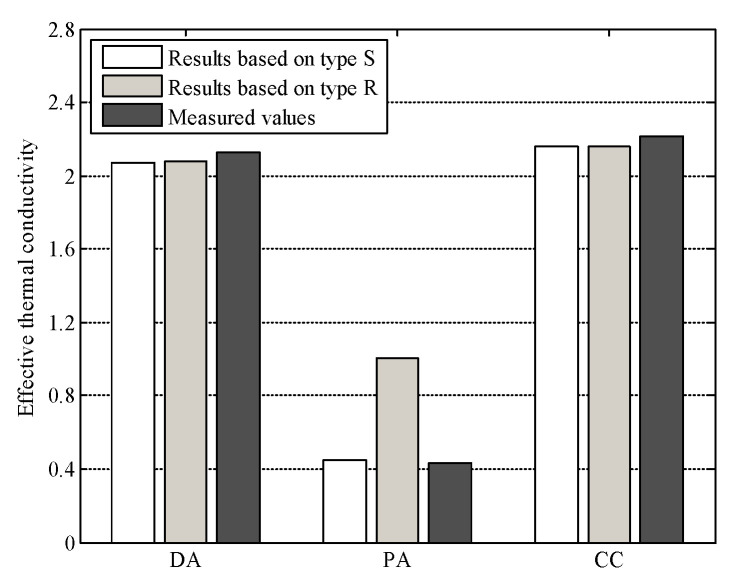
Effective thermal conductivity based on different types of structures.

**Figure 12 materials-16-03322-f012:**
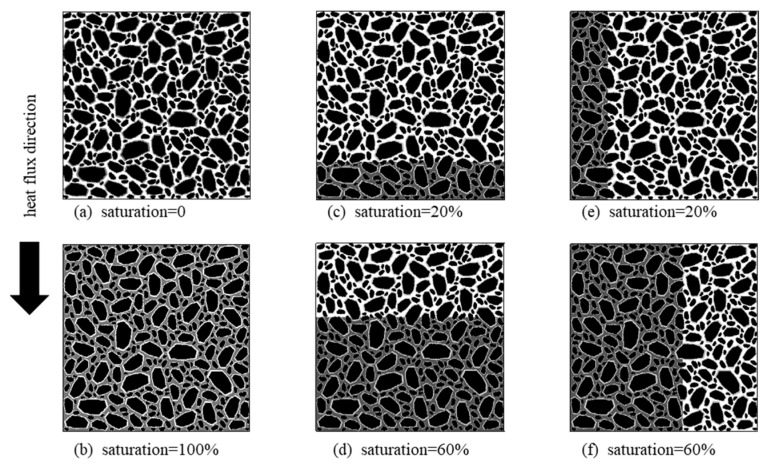
Examples of grouted porous asphalt mixture with different saturations and distributions of grouted materials.

**Figure 13 materials-16-03322-f013:**
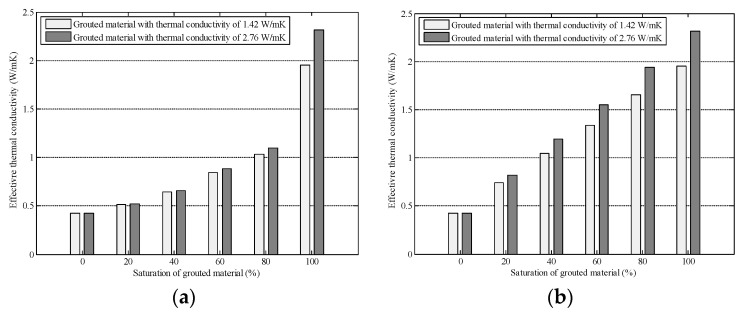
Effective thermal conductivity of grouted porous asphalt mixture with different saturations and distributions of grouted materials (**a**) horizontally distributed (**b**) vertically distributed.

**Figure 14 materials-16-03322-f014:**
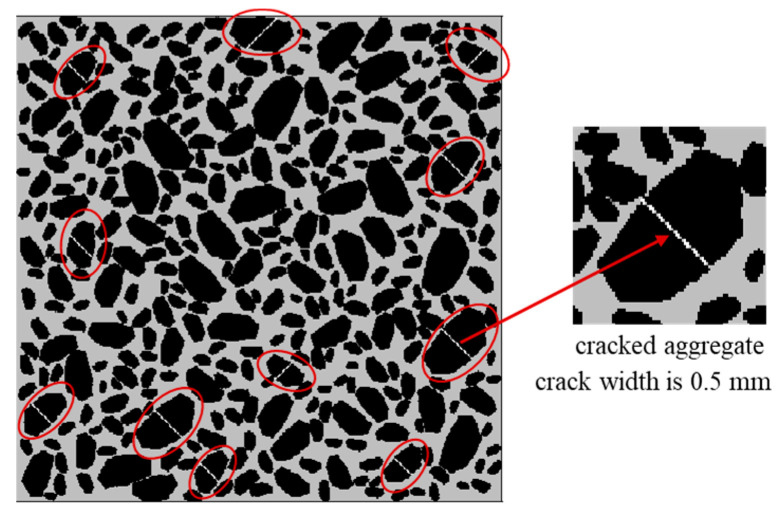
Examples of dense graded asphalt mixture with cracked aggregates.

**Figure 15 materials-16-03322-f015:**
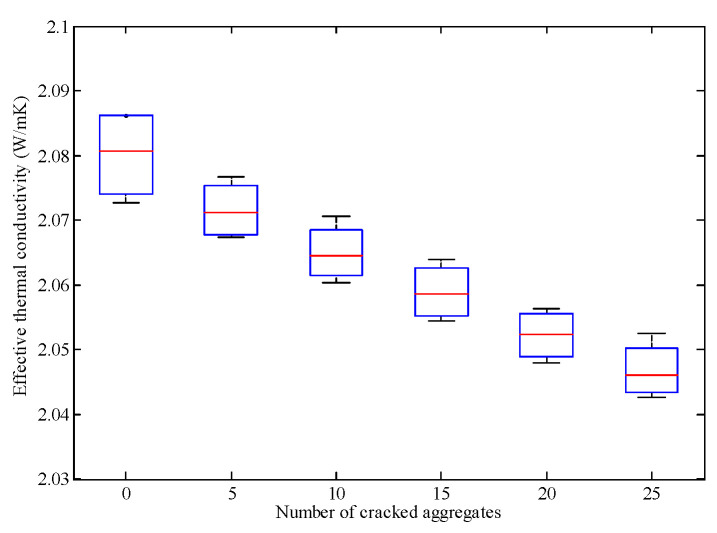
Effect of aggregate cracking on thermal conductivity.

**Figure 16 materials-16-03322-f016:**
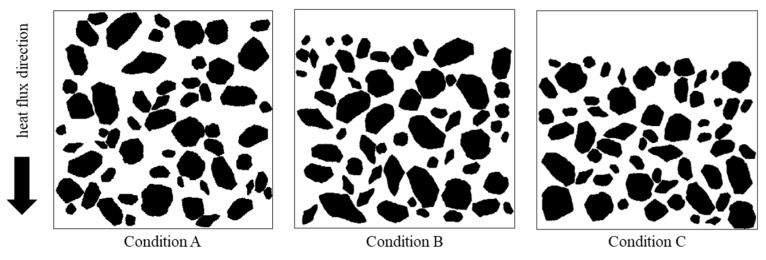
Examples of cement concrete in different segregation conditions.

**Figure 17 materials-16-03322-f017:**
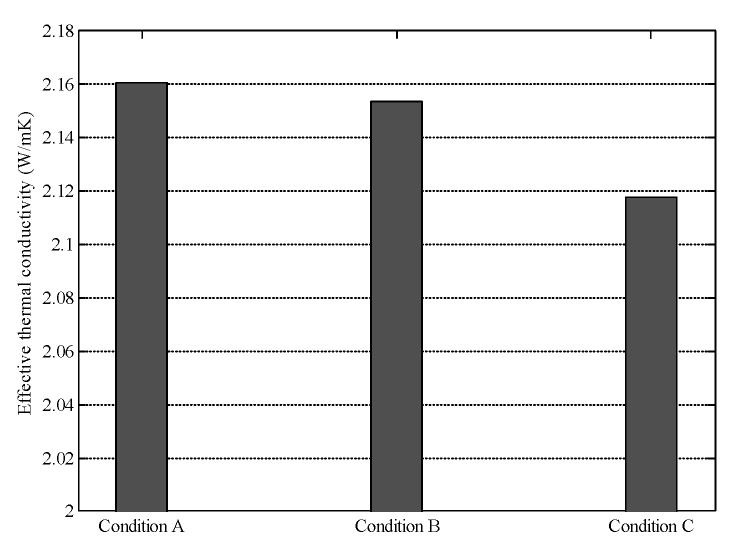
Effect of concrete segregation on thermal conductivity.

**Table 1 materials-16-03322-t001:** Aggregate gradations for different materials.

Sieve Size (mm)	Passing Percentage (%)		
	Dense Graded Asphalt Mixture	Porous Asphalt Mixture	Cement Concrete
19.5	-	-	100.0
16	100	100.0	-
13.2	96.0	93.1	-
9.5	82.1	61.1	52.6
4.75	52.2	22.0	36.8
2.36	30.9	13.2	30.4
1.18	23.0	9.6	21.2
0.6	16.9	8.0	14.8
0.3	11.0	6.9	7.2
0.15	8.4	5.8	4.1
0.075	6.8	5.1	-

**Table 2 materials-16-03322-t002:** Mix proportion of concrete (kg/m^3^).

Cement	Fly Ash	Sand	Gravel	Water	Water Reducer
425	75	662	1127	200	5

**Table 3 materials-16-03322-t003:** Test values of thermal conductivity of various materials (W/mK).

DA	PA	CC	FAM-DA	FAM-PA	FAM-CC
2.127	0.429	2.213	1.322	0.851	1.805

## Data Availability

Data will be made available on request.

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
