# Peer review of "Neural Network Aided Homogenization Approach for Predicting Effective Thermal Conductivity of Composite Construction Materials"

_materials, 2023, doi:10.3390/ma16093322_

Round 1

Reviewer 1 Report

This paper investigated the thermal conductivity of composite construction materials by simulation and experiment methods. The paper provides good-quality writing. However. Several things need to be done :
1. The author did not provide the experiment composite sample's microstructure/SEM image. This is needed to validate the data.
2. (Line 394-395) Please elaborate more on this discussion and add references.
3. Composite conductivity is related to the structure of the sample. If the molecules are close to each other, the conductivity will be high due to the low energy needed when the molecules vibrate. The author needs to add this reference with a deep discussion regarding the mechanism. (https://www.mdpi.com/2073-4360/15/3/643).

Reviewer 2 Report

Manuscript ID: materials-2303454

Title: Neural network aided homogenization approach for predicting effective thermal conductivity of composite construction materials

Journal: materials

Comments to authors:

This paper developed a neural network aided homogenization approach for predicting the effective thermal conductivity of various composite construction materials. The 2-D meso-structures of dense graded asphalt mixture, porous asphalt mixture, and cement concrete were generated and divided into 2n×2n square elements with specific thermal conductivity values. A two-layer feed-forward neural network with sigmoid hidden neurons and linear output neurons was built to predict the effective thermal conductivity of the 2×2 block. By repeatedly using the neural network, the effective thermal conductivities of 2-D meso-structures were calculated. The accuracy of the above neural network aided homogenization approach was validated with experiment, and various factors affecting the effective thermal conductivity were analyzed... Although extensive work has been performed but the novelty can be questioned. However, before it can be considered for publication, please address the following comments for a major revision:

1)      The Abstract should be enriched with the brief details of the methodology. The problem to be addressed in this study should also be highlighted in the Abstract.

2)      Please highlight the novelty in the Abstract also.

3)      A few sentences should be added in the Abstract or Conclusion to state the implications of this study clearly.

4)      The abstract should clearly indicate the relevance of the work for international research.

5)      The authors should also present some quantitative results in the Abstract.

6)      Line 27-29: These sentences can be concisely presented or removed from the Abstract.

7)      English proofreading is required for grammatical mistakes and typos.

8)      The research problem can be better articulated.

9)      The novelty and significance of the present work should be highlighted in the last paragraph of the Introduction section.

10)  The authors are recommended to add latest relevant literature review on such works.

11)  What is the need for this work? Is this work helpful for practical applications? Which applications?

12)  The last part of the introduction should conclude the limitations of the previous studies and provide the main objectives and novelties of this study. You need to clearly address the knowledge gap and provide some meaningful phrases that your study can advance the knowledge and can fill in a knowledge gap that has not been considered yet.

13)  The literature review should be improved by adding latest references and discussion.

14)  Work methodologies need more discussion.

15)  Line 15: Text on the labels of Figure 3 should be enlarged.

16)  Results and Discussion - Here you can discuss your findings, postulate explanations for data, elucidate models, and compare your results with those of others. Be complete but concise. Avoid irrelevant comparisons or contrasts, speculations unsupported by the new information presented in the paper, and verbose discussion.

17)  Results section should be defended using technical reasons and relevant references.

18)  The sources of data and the quality need to be discussed in detail.

19)  Various statistical indices can be added and discussed for the predictions of the proposed ANN models such as R2, RMSE, MAE, MSE, COV, etc…

20)  More technical discussion to the presented experimental results should be added.

21)  There are no critical review/discussions before the Conclusions. Authors should add it.

22)  What is the relevance and importance of the findings found in the present study? It is very important to highlight these aspects.

23)  Conclusions should be refined and briefly presented. More numerical results should be added. The conclusions should be more conclusive and based on the critical analysis of the author.

24)  The discussion and conclusions sections must clearly establish a strong correlation.

25)  Conclusions must go deeper; it would be more interesting if the authors focus more on the significance of their findings, the importance of the study. The barriers to do it, what would be the consequences, in the real world.

26)  What are the limitations of the present study? Please mention them in the manuscript.

27)  The authors can add the future recommendations based on the present study.

Reviewer 3 Report

The thermal conductivity of both concrete and asphalt concrete road pavements was studied in the work. The property is very important, and it should also be taken into account when designing the structural layers of the road according to the MEPDG. However, perhaps it could be emphasized more precisely why this characteristic is very important. One is the thermal conductivity of the “fresh” road pavement, but the thermal conductivity during operation/performance is important, which can be the basis or an important indicator for non-destructive testing of the road pavement and timely identification of damages. Because the thermal conductivity of a damaged pavement (microcracks) will differ from an undamaged one.

Neural model development, homogenization approach and validation should be mentioned as an important innovative and practical value of this work

The question is that usually void content in dense graded asphalt concrete is 2-5%. Why exactly 6%? Also it is recommended even more numerical values ​​in the conclusions.

Thus, in the introductory part, a few sentences would be desirable on what asphalt concrete factors influence  thermal conductivity, such as porosity/voids, amount of bitumen, properties and origin of mineral material, etc. When asphalt/bitumen ages, its properties change drastically, including thermal conductivity. Therefore, it should be emphasized that the characteristics change over time.

In order to include the environmental impact (sustainability) of this approach in the article, it is recommended referring to the current world trend - asphalt concrete from 100% recycled material, for example, DOI: 10.1016/j.conbuildmat.2021.126026

Round 2

Reviewer 1 Report

All of the comments have been addressed. I accept publication in this current form

Reviewer 2 Report

This work can be accepted in its present form.